# Understanding the Generalization Benefit of Normalization Layers: Sharpness Reduction

**Kaifeng Lyu**    **Zhiyuan Li**    **Sanjeev Arora**
Department of Computer Science
Princeton University
{klyu,zhiyuanli,arora}@cs.princeton.edu

## Abstract

Normalization layers (e.g., Batch Normalization, Layer Normalization) were introduced to help with optimization difficulties in very deep nets, but they clearly also help generalization, even in not-so-deep nets. Motivated by the long-held belief that flatter minima lead to better generalization, this paper gives mathematical analysis and supporting experiments suggesting that normalization (together with accompanying weight-decay) encourages GD to reduce the sharpness of loss surface. Here "sharpness" is carefully defined given that the loss is scale-invariant, a known consequence of normalization. Specifically, for a fairly broad class of neural nets with normalization, our theory explains how GD with a finite learning rate enters the so-called Edge of Stability (EoS) regime, and characterizes the trajectory of GD in this regime via a continuous sharpness-reduction flow.

## 1   Introduction

Training modern deep neural nets crucially relies on normalization layers to make the training process less sensitive to hyperparameters and initialization. The two of the most popular normalization layers are Batch Normalization (BN) [55] for vision tasks and Layer Normalization (LN) [9] for language tasks. Recent works also proposed other normalization layers aiming for better performance, most notably including Group Normalization (GN) [120], Weight Normalization (WN) [102], Scaled Weight Standardization (SWS) [97, 53, 14], etc. Most normalization layers amount to a reparametrization of the neural net so that the loss becomes invariant to the scale of most parameters, and with a minor change, to *all* parameters: $\mathcal{L}(c\boldsymbol{w}) = \mathcal{L}(\boldsymbol{w})$ for all scalings $c > 0$ [55, 7, 77]. The current paper assumes this scale-invariance for all parameters and analyzes the trajectory of gradient descent with *weight decay* (WD):

$$\boldsymbol{w}_{t+1} \leftarrow (1 - \hat{\eta}\hat{\lambda})\boldsymbol{w}_t - \hat{\eta}\nabla\mathcal{L}(\boldsymbol{w}_t). \tag{1}$$

The use of WD is a common practice that has been adopted in training state-of-the-art neural nets, such as ResNets [46, 47] and Transformers [29, 15]. Previous ablation studies showed that adding WD to normalized nets indeed leads to better generalization [126, 72, 125]. More notably, Liu et al. [83] conducted experiments of training ResNets initialized from global minima with poor test accuracy, and showed that SGD with WD escapes from those bad global minima and attains good test accuracy. In contrast, training with vanilla SGD yields significant generalization degradation.

In the traditional view, WD regularizes the model by penalizing the parameter norm, but this may appear nonsensical for scale-invariant loss because one can scale down the norm arbitrarily without changing the loss value. However, the scale of the parameter *does* matter in backward propagation, and thus WD can affect the training dynamics. In particular, simple calculus shows $\nabla\mathcal{L}(\boldsymbol{w}) = \frac{1}{\|\boldsymbol{w}\|_2}\nabla\mathcal{L}(\frac{\boldsymbol{w}}{\|\boldsymbol{w}\|_2}) \propto \frac{1}{\|\boldsymbol{w}\|_2}$ and $\nabla^2\mathcal{L}(\boldsymbol{w}) = \frac{1}{\|\boldsymbol{w}\|_2^2}\nabla^2\mathcal{L}(\frac{\boldsymbol{w}}{\|\boldsymbol{w}\|_2}) \propto \frac{1}{\|\boldsymbol{w}\|_2^2}$, so WD is in effect trying to enlarge the gradient and Hessian in training. This makes the training dynamics very different from unnormalized nets and requires revisiting classical convergence analyses [77, 78, 84, 80].

36th Conference on Neural Information Processing Systems (NeurIPS 2022).

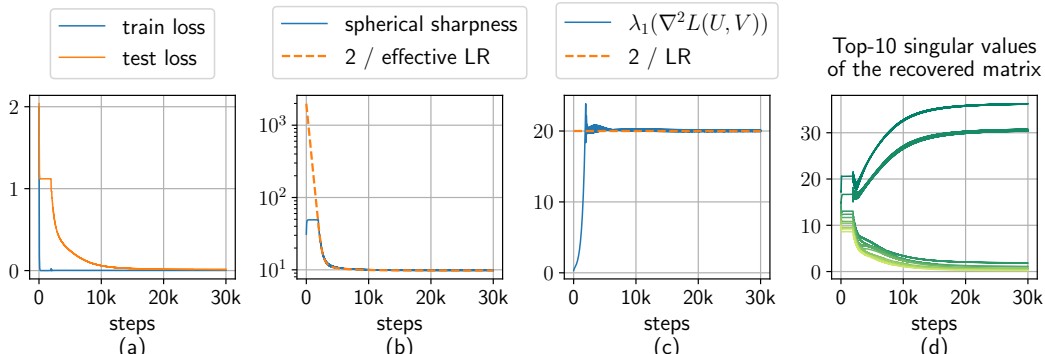

Figure 1: Experiment on overparameterized matrix completion with Batch Normalization. Given 800 (32%) entries $\Omega$ of a rank-2 matrix $\boldsymbol{M} \in \mathbb{R}^{50 \times 50}$, use GD+WD to optimize the loss $\mathcal{L}(\boldsymbol{U}, \boldsymbol{V}) := \frac{1}{|\Omega|} \sum_{(i,j) \in \Omega} (\mathrm{BN}([\boldsymbol{U}\boldsymbol{V}^\top]_{i,j}) - M_{i,j})^2$, where $\boldsymbol{U}, \boldsymbol{V} \in \mathbb{R}^{50 \times 50}$ (thus no explicit constraint on rank). Starting from step $\sim$ 2k, spherical sharpness drops significantly (**b**), which encourages low-rank (**d**) and causes the test loss (MSE of all entries) to decrease from 1.12 to 0.013 (**a**). See also Appendix P.1.

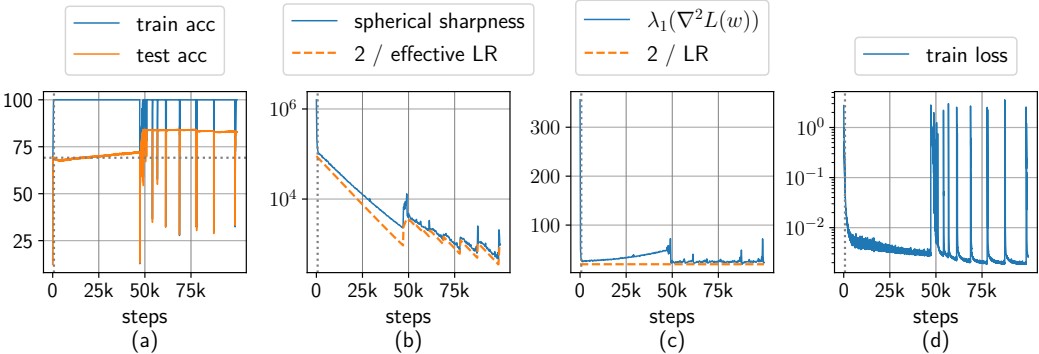

Figure 2: In training a smooth and scale-invariant VGG-11 on CIFAR-10 with (full-batch) GD+WD, the spherical sharpness keeps decreasing and the test accuracy keeps increasing. BN is added after every linear layer to ensure scale-invariance. 100% training accuracy is achieved after $\sim$ 680 steps (dotted line), but as the training continues for $\sim$ 47k steps, the spherical sharpness keeps decreasing (**b**) and the test accuracy increases from 69.1% to 72.0% (**a**). Then the training exhibits destabilization but the test accuracy is further boosted to 84.3%. Removing either of BN or WD eliminates this phenomenon; see Appendices P.4 and P.5.

The current paper aims to improve mathematical understanding of how normalization improves generalization. While this may arise from many places, we focus on studying the dynamics of (full-batch) GD (1), which is a necessary first step towards understanding SGD. We show that the interplay between normalization and WD provably induces an implicit bias to persistently reduce the *sharpness* of the local loss landscape during the training process, which we call the *sharpness-reduction bias*.

It is long believed that flatter minima generalize better [50, 63, 95], but the notion of sharpness/flatness makes sense only if it is carefully defined in consideration of various symmetries in neural nets. One of the most straightforward measures of sharpness is the maximum eigenvalue of Hessian, namely $\lambda_1(\nabla^2 \mathcal{L}(\boldsymbol{w}_t))$. But for normalized nets, this sharpness measure is vulnerable to weight rescaling, because one can scale the weight norm to make a minimizer arbitrarily flat [31]. Also, this sharpness measure may not decrease with the number of training steps: an empirical study by Cohen et al. [24] shows that for various neural nets (including normalized nets), GD has an overwhelming tendency to persistently increase $\lambda_1(\nabla^2 \mathcal{L}(\boldsymbol{w}_t))$ until it reaches the *Edge of Stability (EoS) regime*, a regime where $\lambda_1(\nabla^2 \mathcal{L}(\boldsymbol{w}_t))$ stays around $2/\hat{\eta}$ ($\hat{\eta}$ is the learning rate). See also Section 6 and Figure 2c.

## 1.1 Our Contributions

The sharpness measure we use in this paper takes care of the scale-invariance in normalized nets. We are motivated by our experiments on matrix completion (with BN) and CIFAR-10, where our sharpness measure decreases as the training proceeds, and the generalization improves accordingly; see Figures 1 and 2. We note that techniques from previous works [92, 95, 37] can be easily adopted here to establish a PAC-Bayes bound on the test error, where our sharpness measure appears as an additive term (see Appendix C).

**Definition 1.1** (Spherical Sharpness). For a scale-invariant loss $\mathcal{L}(\boldsymbol{w})$ (i.e., $\mathcal{L}(c\boldsymbol{w}) = \mathcal{L}(\boldsymbol{w})$ for all $c > 0$), the spherical sharpness at $\boldsymbol{w} \in \mathbb{R}^D$ is defined by $\lambda_1(\nabla^2\mathcal{L}(\frac{\boldsymbol{w}}{\|\boldsymbol{w}\|_2}))$, the maximum eigenvalue of the Hessian matrix after projecting $\boldsymbol{w}$ onto the unit sphere.

Based on Definition 1.1, we study the aforementioned sharpness-reduction bias in training normalized nets with GD+WD (defined in (1)). For constant learning rate $\hat{\eta}$ and weight decay $\hat{\lambda}$, we can rewrite this rule equivalently as Projected Gradient Descent (PGD) on the unit sphere with *adaptive* learning rates, $\boldsymbol{\theta}_{t+1} \leftarrow \Pi(\boldsymbol{\theta}_t - \tilde{\eta}_t \nabla\mathcal{L}(\boldsymbol{\theta}_t))$, where $\boldsymbol{\theta}_t := \frac{\boldsymbol{w}_t}{\|\boldsymbol{w}_t\|_2}$ is the direction of $\boldsymbol{w}_t$, and $\tilde{\eta}_t$ is the "effective" learning rate at step $t$ (see Lemma 3.1). We call $\tilde{\eta}_t$ adaptive because it can be shown to resemble the behaviors of adaptive gradient methods (e.g., RMSprop [49]): $\tilde{\eta}_t$ increases when gradient is small and decreases when gradient is large (Figure 3). Our main contributions are as follows:

1. After $\boldsymbol{\theta}_t$ reaches a point near the manifold of minimizers of $\mathcal{L}$, we theoretically show that the effective learning rate $\tilde{\eta}_t$ increases until GD enters a regime where $2/\tilde{\eta}_t$ roughly equals to the spherical sharpness (or equivalently $2/\hat{\eta} \approx \lambda_1(\nabla^2\mathcal{L}(\boldsymbol{w}_t))$), namely the EoS regime (Section 4.1).

2. In the EoS regime, we show that for GD with a small (but finite) learning rate, $\boldsymbol{\theta}_t$ oscillates around the manifold and moves approximately along a sharpness-reduction flow, which is a gradient flow for minimizing spherical sharpness on the manifold (with gradient-dependent learning rate) (Section 4.2).

3. As an application of our theory, we show that for linear regression with BN, GD+WD finds the minimizer that corresponds to the linear model with minimum weight norm, which looks surprisingly the same as the conventional effect of WD but is achieved through the completely different sharpness-reduction mechanism (Section 5).

4. We experimentally verified the sharpness-reduction phenomenon predicted by our theorem and its benefits to generalization on CIFAR-10 with VGG-11 and ResNet-20, as well as matrix completion with BN (Appendix P).

5. We generalize our theoretical results of sharpness-reduction bias to a broader class of adaptive gradient methods, most notably a variant of RMSprop with scalar learning rate (Appendix B).

**Technical Contribution.** Our proof technique is novel and may have independent interest to the ML community. The main challenge is that we need to analyze the implicit bias of GD in the EoS regime which crucially relies on step size being finite — this is in sharp contrast to many previous works on implicit bias of GD [107, 106, 87, 59, 43, 42, 76, 100, 4, 22, 79, 88, 101, 108, 38] where the same bias exists at infinitesimal LR. Our analysis is inspired by a previous line of works [13, 25, 81] showing that label noise can drive SGD to move on the minimizer manifold along the direction of minimizing the trace of Hessian. We borrow a few lemmas from those analyses, but the overall proof strategy is very different because our setting does not even have any stochastic gradient noise. Instead, we connect the dynamics in the EoS regime to power methods and show that GD oscillates around the minimizer manifold. This oscillation then becomes a driving power that pushes the parameter to move on the manifold. Finally, we analyze the speed of this movement by modeling two key parameters of the dynamics as a 1-dimensional Hamiltonian system (Figure 6). To the best of our knowledge, we are the first to provide theoretical proof for a sharpness measure to decrease during the standard GD training, without any additional regularization (e.g., label noise [13, 25, 81]) and without involving uncommon variants of GD (e.g., normalized $\hat{\eta}$ or non-smooth wrappings on the loss function [8]).

## 2 Related Works

**Sharpness and Generalization.** It has been long believed that flat minima generalize better [50]. Several empirical studies [63, 74, 117, 57] verified the positive correlation between flatness and generalization. Neyshabur et al. [95] justified this via PAC-Bayes theory [92]. Several other theoretical papers explored the generalization properties of flat minima specifically for two-layer nets [13, 94, 44, 81, 30] and deep linear nets [93]. Jiang et al. [60] conducted extensive experiments for all existing generalization measures to evaluate their correlation and causal relationships with generalization error, concluding that sharpness-based measures perform the best overall. In light of this, Foret et al. [37] proposed SAM algorithm to improve the generalization by minimizing the sharpness. Despite so many positive results on sharpness-based measures, a common issue of many works is that the measures may suffer from sensitivity to rescaling of parameters in deep nets [31]. Another issue is that the minima could lie in asymmetric valleys that are flat on one side and sharp on the other [45].

**Understanding Normalization Layers.** The benefits of normalization layers can be shown in various aspects. A series of works studied the forward propagation of deep nets at random initialization, showing that normalization layers stabilize the growth of intermediate layer outputs with depth [14, 10, 28], provably avoid rank collapse [26] and orthogonalize representations [27]. Although these works mainly focused on BN [55], Lubana et al. [85], Labatie et al. [67] provided thorough discussions on the applicability of these arguments to other normalization layers. It is also believed that BN has a unique regularization effect through the noise in batch statistics [86, 111, 104]. Several other works argued that normalization layers lead to a smoothening or preconditioning effect of the loss landscape [103, 12, 39, 61, 82, 68], which may help optimization. By analyzing the training dynamics, Arora et al. [7] rigorously proved that normalization yields an auto-tuning effect of the effective learning rate $\tilde{\eta}_t$, which makes the asymptotic speed of optimization much less sensitive to the learning rate and initialization. In linear regression settings, Cai et al. [16], Kohler et al. [65] showed that training with BN leads to a faster convergence rate; Wu et al. [119] studied the implicit regularization effect of WN [102]. For two-layer nets with normalization, Ma and Ying [90] derived a mean-field formulation of the training dynamics; Dukler et al. [33] proved a convergence rate via NTK-based analysis. The current paper focuses on the interplay between normalization and WD during training, whereas all the above works either do not analyze the dynamics or assume no WD.

**Interplay Between Normalization and WD.** A common feature of normalization layers (including but not limited to BN, WN, LN, GN, SWS) is that they make the loss invariant to the scale of layer weights. In presence of both scale-invariance and WD, training dynamics can go out of the scope of the classical optimization theory, e.g., one can train the net to small loss even with learning rates exponentially increasing [77]. A series of works investigated into the interplay between normalization and WD and argued that the training dynamic with SGD eventually reaches an "equilibrium" state, where the parameter norm [78, 113, 21] and the size of angular update [114] become stable. Li et al. [78], Wang and Wang [115] provided empirical and theoretical evidence that the function represented by the net also equilibrates to a stationary distribution that is independent of initialization. This could be related to Liu et al. [83]'s experiments on the ability of SGD with WD to escape from bad initialization, but it remains unclear why the generalization should be good at the equilibrium state. In this paper, we focus on (full-batch) GD, which is the most basic and important special case of SGD.

## 3 Preliminaries

Let $\mathbb{S}^{D-1} := \{\boldsymbol{\theta} \in \mathbb{R}^D : \|\boldsymbol{\theta}\|_2 = 1\}$ be the unit sphere equipped with subspace topology. We say a loss function $\mathcal{L}(\boldsymbol{w})$ defined on $\mathbb{R}^D \setminus \{\mathbf{0}\}$ is *scale-invariant* if $\mathcal{L}(c\boldsymbol{w}) = \mathcal{L}(\boldsymbol{w})$ for all $c > 0$. In other words, the loss value does not change with the parameter norm. For a differentiable scale-invariant function $\mathcal{L}(\boldsymbol{w})$, the gradient is $(-1)$-homogeneous and it is always perpendicular to $\boldsymbol{w}$, i.e., $\nabla\mathcal{L}(c\boldsymbol{w}) = c^{-1}\nabla\mathcal{L}(\boldsymbol{w})$ for all $c > 0$ and $\langle\nabla\mathcal{L}(\boldsymbol{w}), \boldsymbol{w}\rangle = 0$ (see Lemma D.1).

The focus of this paper is the dynamics of GD+WD on scale-invariant loss. (1) gives the update rule for learning rate (LR) $\hat{\eta}$ and weight decay (WD) $\hat{\lambda}$. We use $\boldsymbol{\theta}_t := \frac{\boldsymbol{w}_t}{\|\boldsymbol{w}_t\|_2}$ to denote the projection of $\boldsymbol{w}_t$ onto $\mathbb{S}^{D-1}$ at step $t$. We write GD+WD on scale-invariant loss as a specific kind of Projected Gradient Descent (PGD) and define the *effective learning rate* to be the LR $\tilde{\eta}_t := \frac{\hat{\eta}}{(1-\hat{\eta}\hat{\lambda})\|\boldsymbol{w}_t\|_2^2}$ that appears in the update rule of PGD. This notion is slightly different from the effective learning rate $\frac{\hat{\eta}}{\|\boldsymbol{w}_t\|_2^2}$ defined in previous works [113, 52, 7], but ours is more convenient for our analysis.

**Lemma 3.1.** *When the parameters $\boldsymbol{w}_t$ are updated as* (1), $\boldsymbol{\theta}_t$ *satisfies the following equation:*

$$\boldsymbol{\theta}_{t+1} = \Pi(\boldsymbol{\theta}_t - \tilde{\eta}_t \nabla\mathcal{L}(\boldsymbol{\theta}_t)), \tag{2}$$

*where $\tilde{\eta}_t := \frac{\hat{\eta}}{(1-\hat{\eta}\hat{\lambda})\|\boldsymbol{w}_t\|_2^2}$ is called the **effective learning rate** at step $t$, and $\Pi : \boldsymbol{w} \mapsto \frac{\boldsymbol{w}}{\|\boldsymbol{w}\|_2}$ is the projection operator that projects any vector onto the unit sphere.*

## 4 GD+WD on Scale-Invariant Loss Functions

This section analyzes GD+WD (1) on a scale-invariant loss $\mathcal{L}(\boldsymbol{w})$, in particular what happens after approaching a manifold of local minimizers. Section 4.1 analyzes the dynamics in the stable regime, where loss is guaranteed to decrease monotonically, and Theorem 4.2 suggests $\boldsymbol{w}_t$ can get close to a local minimizer at some time $t_0$. We show that the effective LR keeps increasing after $t_0$, causing

GD+WD to eventually leave this stable regime and enter a new regime which we call the Edge of Stability (EoS). In Section 4.2, we establish our main theorem, which connects the dynamics of $\boldsymbol{w}_t$ in the EoS regime to a sharpness-reduction flow.

## 4.1 GD+WD Eventually Leaves the Stable Regime

A standard step of analyzing optimization methods is to do Taylor expansion locally for the loss function, and show that how the optimization method decreases the loss using a *descent lemma*. In our case of scale-invariant loss functions, we use $\boldsymbol{H}(\boldsymbol{w}) := \nabla^2 \mathcal{L}(\boldsymbol{w}) \in \mathbb{R}^{D \times D}$ to denote the Hessian matrix of $\mathcal{L}$ at $\boldsymbol{w} \in \mathbb{R}^D$, and $\lambda_1^{\mathrm{H}}(\boldsymbol{w}) := \lambda_1(\boldsymbol{H}(\boldsymbol{w}))$ to denote the top eigenvalue of $\boldsymbol{H}(\boldsymbol{w})$.

**Lemma 4.1** (Descent Lemma). *For scale-invariant loss $\mathcal{L}(\boldsymbol{w})$, at step $t$ of GD+WD we have*

$$\mathcal{L}(\boldsymbol{\theta}_{t+1}) \leq \mathcal{L}(\boldsymbol{\theta}_t) - \tilde{\eta}_t(1 - \tilde{\eta}_t \lambda_{\max}^{(t)}/2) \|\nabla \mathcal{L}(\boldsymbol{\theta}_t)\|_2^2.$$

*where $\lambda_{\max}^{(t)} := \sup_{\alpha \in [0, \tilde{\eta}_t]} \left\{ \lambda_1^{\mathrm{H}}(\boldsymbol{\theta}_t - \alpha \nabla \mathcal{L}(\boldsymbol{\theta}_t)) \right\}$ is an upper bound of spherical sharpness locally.*

This descent lemma shows that the training loss $\mathcal{L}(\boldsymbol{\theta}_t)$ keeps decreasing as long as the effective LR $\tilde{\eta}_t$ is smaller than $2/\lambda_{\max}^{(t)}$, We call the regime of $\tilde{\eta}_t < 2/\lambda_{\max}^{(t)}$ as the *stable regime* of GD+WD. If $\tilde{\eta}_t \approx 2/\lambda_{\max}^{(t)}$ with a small difference, then we call it as the *Edge of Stability (EoS) regime*. We remark that this condition for EoS regime is essentially the *same* as $\hat{\eta} \approx 2/\lambda_1^{\mathrm{H}}(\boldsymbol{w})$ in Cohen et al. [24]'s definition because $\tilde{\eta}_t \cdot \lambda_{\max}^{(t)} \approx \hat{\eta} \cdot \lambda_1^{\mathrm{H}}(\boldsymbol{w})$; see Appendix G.3.

Fix an initial point $\boldsymbol{w}_0 \in \mathbb{R}^D \setminus \{\boldsymbol{0}\}$. Now we aim to characterize the dynamics of GD+WD when LR $\hat{\eta}$ and WD $\hat{\lambda}$ are small enough. The convergence rate of GD+WD has been analyzed by Li et al. [80]. Here we present a variant of their theorem that bounds both the gradient and effective LR.

**Theorem 4.2** (Variant of Theorem D.2, Li et al. [80]). *Let $\mathcal{L}(\boldsymbol{w})$ be a scale-invariant loss function and $\rho_2 := \sup\{\|\nabla^2 \mathcal{L}(\boldsymbol{w})\|_2 : \boldsymbol{w} \in \mathbb{S}^{D-1}\}$ be the smoothness constant of $\mathcal{L}$ restricted on the unit sphere. For GD+WD (1) with $\hat{\eta}\hat{\lambda} \leq 1/2$ and $\tilde{\eta}_0 \leq \frac{1}{\pi^2 \rho_2(1-\hat{\eta}\hat{\lambda})}$, let $T_0 := \left\lceil \frac{1}{2\hat{\eta}\hat{\lambda}} \ln \frac{\|\boldsymbol{w}_0\|_2^2}{\rho_2 \pi^2 \hat{\eta}} \right\rceil$ steps, there must exist $0 \leq t \leq T_0$ such that $\|\nabla \mathcal{L}(\boldsymbol{\theta}_t)\|_2^2 \leq 8\pi^4 \rho_2^2 \hat{\lambda}\hat{\eta}$ and $\tilde{\eta}_t \leq \frac{2}{\pi^2 \rho_2(1-\hat{\eta}\hat{\lambda})}$.*

Theorem 4.2 shows that for some $t_0 \leq T_0$, $\|\nabla \mathcal{L}(\boldsymbol{\theta}_{t_0})\|_2^2 \leq O(\hat{\lambda}\hat{\eta})$ and $\tilde{\eta}_{t_0} \leq \frac{1}{\pi^2 \rho_2} < \frac{2}{\rho_2}$, which means $\boldsymbol{\theta}_{t_0}$ is an approximate first-order stationary point of $\mathcal{L}$ on the unit sphere. This does not guarantee that $\boldsymbol{\theta}_{t_0}$ is close to any global minimizer, but in practice the training loss rarely gets stuck at a non-optimal value when the model is overparameterized [70, 96, 71, 125]. We are thus motivated to study the case where $\boldsymbol{\theta}_{t_0}$ not only has small gradient $\|\nabla \mathcal{L}(\boldsymbol{\theta}_{t_0})\|_2^2 \leq O(\hat{\lambda}\hat{\eta})$ but also is close to a local minimizer $\boldsymbol{\theta}^* \in \mathbb{S}^{D-1}$ of $\mathcal{L}$ in the sense that $\|\boldsymbol{\theta}_{t_0} - \boldsymbol{\theta}^*\|_2 \leq O((\hat{\lambda}\hat{\eta})^{1/2})$ (assuming smoothness, the latter implies the former).

As the gradient is small near the local minimizer $\boldsymbol{\theta}^*$, starting from step $t_0$, the norm of $\boldsymbol{w}_t$ decreases due to the effect of WD. See Figure 3a. Since the effective LR is inversely proportional to $\|\boldsymbol{w}_t\|_2^2$, this leads to the effective LR to increase. Then Theorem 4.4 will show that the GD+WD dynamic eventually leaves the stable regime at some time $t_1 > t_0$, and enters the EoS regime where $\tilde{\eta}_t \approx 2/\lambda_{\max}^{(t)}$.

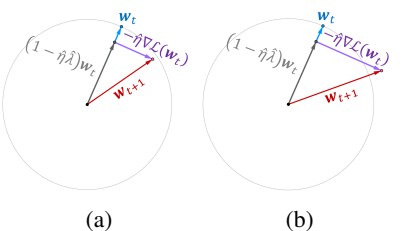

(a)          (b)

Figure 3: The norm of $\boldsymbol{w}_t$ decreases when gradient is small and increases when gradient is large.

To establish Theorem 4.4, we need to assume that $\mathcal{L}$ satisfies Polyak-Łojasiewicz (PL) condition locally, which is a standard regularity condition in the optimization literature to ease theoretical analysis around a minimizer. Intuitively, PL condition guarantees that the gradient grows faster than a quadratic function as we move a parameter $\boldsymbol{\theta}$ away from $\boldsymbol{\theta}^*$. Note that PL condition is strictly weaker than convexity as the function can still be non-convex under PL condition (see, e.g., [62]).

**Definition 4.3** (Polyak-Łojasiewicz Condition). For a scale-invariant loss $\mathcal{L}(\boldsymbol{w})$ and $\mu > 0$, we say that $\mathcal{L}$ satisfies $\mu$-Polyak-Łojasiewicz condition (or $\mu$-PL) locally around a local minimizer $\boldsymbol{\theta}^*$ on $\mathbb{S}^{D-1}$ if for some neighborhood $U \subseteq \mathbb{S}^{D-1}$ of $\boldsymbol{\theta}^*$, $\forall \boldsymbol{\theta} \in U : \frac{1}{2}\|\nabla \mathcal{L}(\boldsymbol{\theta})\|_2^2 \geq \mu \cdot (\mathcal{L}(\boldsymbol{\theta}) - \mathcal{L}(\boldsymbol{\theta}^*))$.

**Theorem 4.4.** *Let $\mathcal{L}(\boldsymbol{w})$ be a $\mathcal{C}^2$-smooth scale-invariant loss that satisfies $\mu$-PL around a local minimizer $\boldsymbol{\theta}^*$ on the unit sphere, and $\rho_2 := \sup\{\|\nabla^2 \mathcal{L}(\boldsymbol{w})\|_2 : \boldsymbol{w} \in \mathbb{S}^{D-1}\}$. For GD+WD on $\mathcal{L}(\boldsymbol{w})$ with learning rate $\hat{\eta}$ and weight decay $\hat{\lambda}$, if at some step $t_0$, $\|\boldsymbol{\theta}_{t_0} - \boldsymbol{\theta}^*\|_2 \leq O((\hat{\lambda}\hat{\eta})^{1/2})$ and $\tilde{\eta}_{t_0} \leq \frac{2}{\rho_2} < \frac{2}{\lambda_1^{\mathrm{H}}(\boldsymbol{\theta}^*)}$, and if $\hat{\lambda}\hat{\eta}$ is small enough, then there exists a time $t_1 > t_0$ such that $\|\boldsymbol{\theta}_{t_1} - \boldsymbol{\theta}^*\|_2 = O((\hat{\lambda}\hat{\eta})^{1/2})$ and $\tilde{\eta}_{t_1} = \frac{2}{\lambda_1^{\mathrm{H}}(\boldsymbol{\theta}^*)} + O((\hat{\lambda}\hat{\eta})^{1/2})$.*

## 4.2 Dynamics at the Edge of Stability

From the analysis in the previous subsection, we know that $\boldsymbol{\theta}_t$ can get close to a local minimizer $\boldsymbol{\theta}^*$ and enter the EoS regime at some step $t_1$. But what happens after $t_1$?

Figure 4 gives a warm-up example on a 3D scale-invariant loss $\mathcal{L} : \mathbb{R}^3 \setminus \{\mathbf{0}\} \to \mathbb{R}$, where the black line is a manifold $\Gamma$ consisting of all the minimizers. In training with GD+WD, $\boldsymbol{\theta}_t$ first goes close to a local minimizer $\boldsymbol{\zeta}_0$, then Theorem 4.4 suggests that WD causes the effective LR to steadily increase until the dynamic enters the EoS regime. Now something interesting happens — $\boldsymbol{\theta}_t$ moves a bit away from $\boldsymbol{\zeta}_0$ and starts to oscillate around the manifold $\Gamma$. This oscillation is not completely perpendicular to $\Gamma$ but actually forms a small angle that pushes $\boldsymbol{\theta}_t$ to move downward persistently until $\boldsymbol{\theta}_t$ approaches the minimizer $\boldsymbol{\zeta}_*$ denoted in the plot.

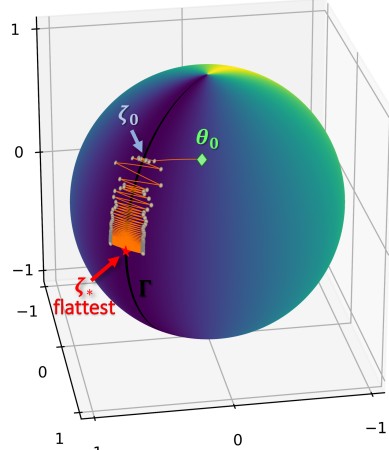

Figure 4: The trajectory of $\boldsymbol{\theta}_t$ on a 3D scale-invariant loss function. Darker color means lower loss on the unit sphere, and points in the black line are minimizers (see Appendix F). In the end, $\boldsymbol{\theta}_t$ approaches the flattest one (red star).

For a general scale-invariant loss $\mathcal{L} : \mathbb{R}^D \setminus \{\mathbf{0}\} \to \mathbb{R}$, which minimizer does $\boldsymbol{\theta}_t$ move towards? In this work, we consider the setting where there is a manifold $\Gamma$ consisting only of local minimizers (but not necessarily all of them). We show that $\boldsymbol{\theta}_t$ always oscillates around the manifold once it approaches the manifold and enters the EoS regime, and meanwhile $\boldsymbol{\theta}_t$ keeps moving in a direction of reducing spherical sharpness.

### 4.2.1 Assumptions

Now we formally introduce our main assumption on the local minimizer manifold $\Gamma$.

**Assumption 4.5.** *The loss function $\mathcal{L} : \mathbb{R}^D \setminus \{\mathbf{0}\} \to \mathbb{R}$ is $\mathcal{C}^4$-smooth and scale-invariant. $\Gamma$ is a $\mathcal{C}^2$-smooth, $(D_\Gamma - 1)$-dimensional submanifold of $\mathbb{S}^{D-1}$ for some $0 \leq D_\Gamma < D$, where every $\boldsymbol{\theta} \in \Gamma$ is a local minimizer of $\mathcal{L}$ on $\mathbb{S}^{D-1}$ and $\mathrm{rank}(\boldsymbol{H}(\boldsymbol{\theta})) = D - D_\Gamma$.*

Scale-invariance has become a standard assumption in studying neural nets with normalization layers [77, 78, 84]. For VGG and ResNet, the scale-invariance can be ensured after making minor changes to the architectures (see Appendix Q.1). The training loss $\mathcal{L}$ may not be smooth if the activation is ReLU, but lately it has become clear that differentiable activations such as Swish [98], GeLU [48] can perform equally well. Swish is indeed used in our VGG-11 experiments (Figure 2), but ResNet with ReLU activation also exhibits a sharpness-reduction bias empirically (see Appendix P.2).

For any local minimizer $\boldsymbol{\theta} \in \Gamma$, the eigenvalues $\lambda_k^{\mathrm{H}}(\boldsymbol{\theta})$ must be non-negative. And $\lambda_k^{\mathrm{H}}(\boldsymbol{\theta}) = 0$ for all $D - D_\Gamma < k \leq D$, since $\Gamma$ is of dimension $D_\Gamma - 1$. The condition $\mathrm{rank}(\boldsymbol{H}(\boldsymbol{\theta})) = D - D_\Gamma$ ensures that the Hessian is maximally non-degenerate on $\Gamma$, which also appears as a key assumption in previous works [81, 8, 35]. This condition simplifies the calculus on $\Gamma$ in our analysis as it ensures that the null space of the matrix $\boldsymbol{H}(\boldsymbol{\theta})$ equals to the tangent space of $\Gamma$ at $\boldsymbol{\theta} \in \Gamma$. It is also closely related to PL condition (Definition 4.3) as Assumption 4.5 implies that $\mathcal{L}(\boldsymbol{\theta})$ satisfies $\mu$-PL (for some $\mu > 0$) locally around every $\boldsymbol{\theta} \in \Gamma$ on the unit sphere (Arora et al. [8], Lemma B.3).

To ease our analysis, we also need the following regularity condition to ensure that the largest eigenvalue is unique. In our experiments, sharpness reduction happens even when the multiplicity of the top eigenvalue is more than 1, but we leave the analysis of that case to future work.

**Assumption 4.6.** *For all $\boldsymbol{\theta} \in \Gamma$, $\lambda_1^{\mathrm{H}}(\boldsymbol{\theta}) > \lambda_2^{\mathrm{H}}(\boldsymbol{\theta})$. That is, the top eigenvalue of $\boldsymbol{H}(\boldsymbol{\theta})$ is unique.*

### 4.2.2 Main Theorem

First, we define $\eta_{\text{in}} := \hat{\eta}\hat{\lambda}$ as the intrinsic learning rate (name from Li et al. [78]) for convenience. As suggested in Theorems 4.2 and 4.4, $\boldsymbol{\theta}_t$ can get close to a local minimizer and be in the EoS regime at some step $t_1$: if $\boldsymbol{\zeta}_0$ is the local minimizer, then $\|\boldsymbol{\theta}_{t_1} - \boldsymbol{\zeta}_0\|_2 = O(\eta_{\text{in}}^{1/2})$ and $\tilde{\eta}_{t_1} = \frac{2}{\lambda_1^{\text{H}}(\boldsymbol{\zeta}_0)} + O(\eta_{\text{in}}^{1/2})$. In our main theorem, we start our analysis from step $t_1$ while setting $t_1 = 0$ WLOG (otherwise we can shift the step numbers). We connect GD+WD in the EoS regime to the following gradient flow (3) on the manifold $\Gamma$ minimizing spherical sharpness (with gradient-dependent learning rate), and show that one step of GD+WD tracks a time interval of length $\eta_{\text{in}}$ in the gradient flow.

$$\boldsymbol{\zeta}(0) = \boldsymbol{\zeta}_0 \in \Gamma, \qquad \frac{\mathrm{d}}{\mathrm{d}\tau}\boldsymbol{\zeta}(\tau) = -\frac{2\nabla_\Gamma \log \lambda_1^{\text{H}}(\boldsymbol{\zeta}(\tau))}{4 + \|\nabla_\Gamma \log \lambda_1^{\text{H}}(\boldsymbol{\zeta}(\tau))\|_2^2}. \tag{3}$$

Here we use the notation $\nabla_\Gamma R(\boldsymbol{\theta})$ for any $R : \mathbb{R}^D \to \mathbb{R}$ to denote the projection of $\nabla R(\boldsymbol{\theta})$ onto the tangent space $\mathsf{T}_{\boldsymbol{\theta}}(\Gamma)$ at $\boldsymbol{\theta} \in \Gamma$. $\boldsymbol{\zeta}(\tau)$ reduces sharpness as it moves in direction of the negative gradient of $\log \lambda_1^{\text{H}}(\boldsymbol{\zeta}(\tau))$ on $\Gamma$. A simple chain rule shows how fast the spherical sharpness decreases:

$$\frac{\mathrm{d}}{\mathrm{d}t} \log \lambda_1^{\text{H}}(\boldsymbol{\zeta}(\tau)) = -\frac{2\|\nabla_\Gamma \log \lambda_1^{\text{H}}(\boldsymbol{\zeta}(\tau))\|_2^2}{4 + \|\nabla_\Gamma \log \lambda_1^{\text{H}}(\boldsymbol{\zeta}(\tau))\|_2^2} \approx \begin{cases} -\frac{1}{2}\|\nabla_\Gamma \log \lambda_1^{\text{H}}(\boldsymbol{\zeta}(\tau))\|_2^2 & \text{for small gradient;} \\ -2 & \text{for large gradient.} \end{cases}$$

Note that it is not enough to just assume that $\boldsymbol{\theta}_0$ is close to $\boldsymbol{\zeta}_0$. If $\boldsymbol{\theta}_0 = \boldsymbol{\zeta}_0$ holds exactly, then the subsequent dynamic of $\boldsymbol{w}_t$ is described by $\boldsymbol{w}_t = (1 - \hat{\eta}\hat{\lambda})^t \boldsymbol{w}_0$ with direction unchanged. There are also some other bad initial directions of $\boldsymbol{w}_0$ that may not lead to the sharpness-reduction bias. This motivates us to do a smoothed analysis for the initial direction: the initial direction is $\boldsymbol{\zeta}$ with tiny random perturbation, where the perturbation scale is allowed to vary from $\exp(-\eta_{\text{in}}^{-o(1)})$ to $\eta_{\text{in}}^{1/2-o(1)}$, and we show that a good initial direction is met with high probability as $\eta_{\text{in}} \to 0$.[1] Alternatively, one can regard it as a modeling of the tiny random noise in GD+WD due to the precision errors in floating-point operations. See Figure 5b; the training loss can never be exactly zero in practice.

**Initialization Scheme.** Given a local minimizer $\boldsymbol{\zeta}_0 \in \Gamma$, we initialize $\boldsymbol{w}_0 \in \mathbb{R}^D \setminus \{\boldsymbol{0}\}$ as follows: draw $\boldsymbol{\xi} \sim \mathcal{N}(\boldsymbol{0}, \sigma_0^2 \boldsymbol{I}/D)$ from Gaussian and set the direction of $\boldsymbol{w}_0$ to $\frac{\boldsymbol{\zeta}_0 + \boldsymbol{\xi}}{\|\boldsymbol{\zeta}_0 + \boldsymbol{\xi}\|_2}$, where $\sigma_0$ can take any value in $[\exp(-\eta_{\text{in}}^{-o(1)}), \eta_{\text{in}}^{1/2-o(1)}]$; then set the parameter norm $\|\boldsymbol{w}_0\|_2$ to be any value that satisfies $\left|\tilde{\eta}_0 - \frac{2}{\lambda_1^{\text{H}}(\boldsymbol{\zeta}_0)}\right| \le \eta_{\text{in}}^{1/2-o(1)}$, where $\tilde{\eta}_0 := \frac{\hat{\eta}}{(1-\hat{\eta}\hat{\lambda})\|\boldsymbol{w}_0\|_2^2}$ is the effective LR for the first step.

**Theorem 4.7.** *Under Assumptions 4.5 and 4.6, for GD+WD (1) with sufficiently small intrinsic learning rate $\eta_{\text{in}} := \hat{\eta}\hat{\lambda}$, if we follow the above initialization scheme for some $\boldsymbol{\zeta}_0 \in \Gamma$, then with probability $1 - O(\eta_{\text{in}}^{1/2-o(1)})$, the trajectory of $\boldsymbol{\theta}_t := \frac{\boldsymbol{w}_t}{\|\boldsymbol{w}_t\|_2}$ approximately tracks a sharpness-reduction flow $\boldsymbol{\zeta} : [0, T] \to \Gamma$ that starts from $\boldsymbol{\zeta}_0$ and evolves as the ODE (3) up to time $T$ (if solution exists), in the sense that $\|\boldsymbol{\theta}_t - \boldsymbol{\zeta}(t\eta_{\text{in}})\|_2 = O(\eta_{\text{in}}^{1/4-o(1)})$ for all $0 \le t \le T/\eta_{\text{in}}$.*

**Remark 4.8** (Magnitude of Oscillation). As suggested by Figure 4, $\boldsymbol{\theta}_t$ actually oscillates around the manifold. But according to our analysis, the magnitude of oscillation is as small as $O(\eta_{\text{in}}^{1/2-o(1)})$, so it is absorbed into our final bound $O(\eta_{\text{in}}^{1/4-o(1)})$ for the distance between $\boldsymbol{\theta}_t$ and $\boldsymbol{\zeta}(t\eta_{\text{in}})$.

### 4.2.3 Proof Idea

Throughout our proof, we view GD+WD for $\boldsymbol{w}_t$ as a PGD for $\boldsymbol{\theta}_t$ with effective LR $\tilde{\eta}_t$ (Lemma 3.1). To track $\boldsymbol{\theta}_t$ with $\boldsymbol{\zeta}(t\eta_{\text{in}})$, for each step $t$, we construct a local minimizer $\boldsymbol{\phi}_t \in \Gamma$ that serves as the "projection" of $\boldsymbol{\theta}_t$ onto the manifold $\Gamma$, in the sense that the displacement $\boldsymbol{x}_t := \boldsymbol{\theta}_t - \boldsymbol{\phi}_t$ is approximately perpendicular to the tangent space of $\Gamma$ at $\boldsymbol{\phi}_t$. Our entire proof works through induction. According to the initial conditions, the dynamic is initially in the EoS regime: $\|\boldsymbol{x}_t\|_2 \le \eta_{\text{in}}^{1/2-o(1)}$ and $|\tilde{\eta}_t - 2/\lambda_1^{\text{H}}(\boldsymbol{\phi}_t)| \le \eta_{\text{in}}^{1/2-o(1)}$ at $t = 0$. In our induction, we maintain the induction hypothesis that these two EoS conditions continue to hold for all $t \ge 0$.

---

[1]Here $\eta_{\text{in}}^{-o(1)}$ can be constant, $O(\log(1/\eta_{\text{in}}))$, or $O(\text{polylog}(1/\eta_{\text{in}}))$, but not $\eta_{\text{in}}^{-\epsilon}$ if $\epsilon > 0$ is a constant. As mentioned later, this need for random initialization is very similar to the one needed in power method for computing eigenvalues.

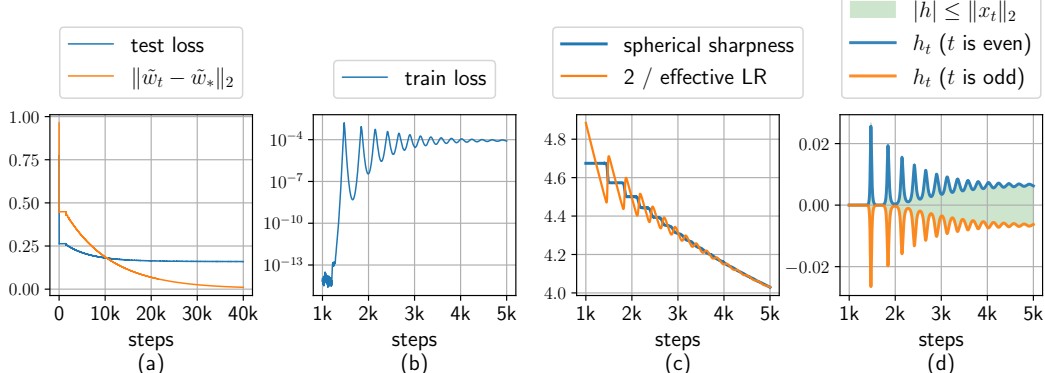

Figure 5: Illustration of the oscillation and periodic behaviors of GD+WD on linear regression with BN (see Sections 4.2.3 and 5). The training loss decreases to $\approx 10^{-14}$ in the first 1k steps and achieves test loss 0.26. Starting from step $\sim 1$k, the dynamic enters the EoS regime. **(a).** The test loss decreases to 0.16 as a distance measure to the flattest solution (M) decreases towards 0; **(b).** The training loss oscillates around $\sim 10^{-4}$ in the EoS regime; **(c).** $2/\tilde{\eta}_t$ switches back and forth between being smaller and larger than $\lambda_1^{\mathrm{H}}(\boldsymbol{\phi}_t)$; **(d).** The parameter oscillates around the minimizer manifold along the top eigenvector direction, and the magnitude of oscillation $|h_t|$ rises and falls periodically.

**Period-Two Oscillation.** A key insight in our proof is that after a few initial steps, $\boldsymbol{\theta}_t$ is oscillating around $\boldsymbol{\phi}_t$ along the $\pm \boldsymbol{v}_1^{\mathrm{H}}(\boldsymbol{\theta})$ directions, where $\boldsymbol{v}_1^{\mathrm{H}}(\boldsymbol{\theta})$ is a unit top eigenvector of $\boldsymbol{H}(\boldsymbol{\theta})$ and is chosen in a way that $\boldsymbol{v}_1^{\mathrm{H}}(\boldsymbol{\theta})$ is continuous on $\Gamma$. More specifically, $\boldsymbol{x}_t = h_t \boldsymbol{v}_1^{\mathrm{H}}(\boldsymbol{\phi}_t) + O(\|\boldsymbol{x}_t\|_2^2)$ for $h_t := \langle \boldsymbol{x}_t, \boldsymbol{v}_1^{\mathrm{H}}(\boldsymbol{\phi}_t) \rangle$. The oscillation is of period 2: $h_t > 0$ when $t$ is even and $h_t < 0$ when $t$ is odd. See Figure 5d for an example.

This oscillation can be connected to a power method for the matrix $\boldsymbol{I} - \tilde{\eta}_t \boldsymbol{H}(\boldsymbol{\phi}_t)$. In the EoS regime, we can approximate $\boldsymbol{\theta}_{t+1}$ (when $\boldsymbol{x}_t$ is small) as $\boldsymbol{\theta}_{t+1} = \Pi(\boldsymbol{\theta}_t - \tilde{\eta}_t \nabla \mathcal{L}(\boldsymbol{\theta}_t)) \approx \Pi(\boldsymbol{\theta}_t - \tilde{\eta}_t \boldsymbol{H}(\boldsymbol{\phi}_t) \boldsymbol{x}_t) \approx \boldsymbol{\theta}_t - \tilde{\eta}_t \boldsymbol{H}(\boldsymbol{\phi}_t) \boldsymbol{x}_t$ by Taylor expansions of $\nabla \mathcal{L}$ and $\Pi : \mathbb{R}^D \setminus \{\boldsymbol{0}\} \to \mathbb{S}^{D-1}$. We can further show that $\boldsymbol{\phi}_{t+1} \approx \boldsymbol{\phi}_t$ due to our choice of projections. Then the connection to power method is shown below:

$$\boldsymbol{x}_{t+1} \approx \boldsymbol{\theta}_{t+1} - \boldsymbol{\phi}_t \approx (\boldsymbol{I} - \tilde{\eta}_t \boldsymbol{H}(\boldsymbol{\phi}_t)) \boldsymbol{x}_t.$$

By simple linear algebra, $\boldsymbol{v}_1^{\mathrm{H}}(\boldsymbol{\phi}_t)$ is an eigenvector of $\boldsymbol{I} - \tilde{\eta}_t \boldsymbol{H}(\boldsymbol{\phi}_t)$, associated with eigenvalue $1 - \tilde{\eta}_t \lambda_1^{\mathrm{H}}(\boldsymbol{\phi}_t) \approx -1$. The remaining eigenvalues are $\{1 - \tilde{\eta}_t \lambda_i^{\mathrm{H}}(\boldsymbol{\phi}_t)\}_{i=2}^D$, where $\lambda_i^{\mathrm{H}}(\boldsymbol{\phi}_t)$ is the $i$-th largest eigenvalue of $\boldsymbol{H}(\boldsymbol{\theta}_t)$, and they lie in the range $(-1, 1]$ since $\lambda_i^{\mathrm{H}}(\boldsymbol{\phi}_t) \in [0, \lambda_1^{\mathrm{H}}(\boldsymbol{\phi}_t))$. Using a similar analysis to power method, we show that $\boldsymbol{x}_t$ quickly aligns to the direction of $\pm \boldsymbol{v}_1^{\mathrm{H}}(\boldsymbol{\phi}_t)$ after a few initial steps, as the corresponding eigenvalue has approximately the largest absolute value.[2]

To formally establish the above result, we need a tiny initial alignment between $\boldsymbol{x}_0$ and $\boldsymbol{v}_1^{\mathrm{H}}(\boldsymbol{\phi}_0)$, just as the initial condition in power method. This is where we need the initial random perturbation.

**Oscillation Drives $\boldsymbol{\phi}_t$ to Move.** This period-two oscillation is the driving power to push $\boldsymbol{\phi}_t$ to move on the manifold. The main idea here is to realize that the oscillation direction deviates slightly from the direction of $\pm \boldsymbol{v}_1^{\mathrm{H}}(\boldsymbol{\phi}_t)$ by using a higher-order approximation. We specifically use the Taylor approximation to show that this deviation leads $\boldsymbol{\phi}_t$ to move slightly on $\Gamma$: after each cycle of oscillation, $\boldsymbol{\phi}_{t+2} \approx \boldsymbol{\phi}_t - 4h_t^2 \nabla_\Gamma \log \lambda_1^{\mathrm{H}}(\boldsymbol{\phi}_t) + O(\eta_{\mathrm{in}}^{1.5 - o(1)})$, which resembles two steps of gradient descent on $\Gamma$ to minimize the logarithm of spherical sharpness with learning rate $2h_t^2$,

**Periodic Behavior of $h_t$ and $\tilde{\eta}_t$.** It remains to analyze the dynamics of $h_t$ so that we can know how fast the sharpness reduction is. Our analysis is inspired by an empirical study from Lobacheva et al. [84], which reveals a periodic behavior of gradients and effective learning rates in training normalized nets with weight decay. In our theoretical setting, we capture this periodic behavior by showing that $h_t$ and $\tilde{\eta}_t$ *do* evolve periodically. See Figures 5c and 5d for an example.

The key is that $\tilde{\eta}_t$ changes as an adaptive gradient method: $\tilde{\eta}_t$ increases when gradient is small and decreases when gradient is large (due to the effect of WD; see Figures 3a and 3b), and in our case the gradient norm scales as $|h_t|$ since $\nabla \mathcal{L}(\boldsymbol{\theta}_t) \approx h_t \lambda_1^{\mathrm{H}}(\boldsymbol{\phi}_t) \boldsymbol{v}_1^{\mathrm{H}}(\boldsymbol{\phi}_t)$. By the power method approximation,

---

[2]Our construction of $\boldsymbol{\phi}_t$ ensures that $\boldsymbol{x}_t$ only has a small overlap with the 1-eigenspace of $\boldsymbol{I} - \tilde{\eta}_t \boldsymbol{H}(\boldsymbol{\phi}_t)$, so $\boldsymbol{x}_t$ can only align to $\pm \boldsymbol{v}_1^{\mathrm{H}}(\boldsymbol{\phi}_t)$.

$h_{t+2} \approx (1 - \tilde{\eta}_t \lambda_1^{\mathrm{H}}(\boldsymbol{\phi}_t))^2 h_t$, so $|h_t|$ decreases when $\tilde{\eta}_t < 2/\lambda_1^{\mathrm{H}}(\boldsymbol{\phi}_t)$. But $|h_t|$ cannot decrease forever, since $\tilde{\eta}_t$ increases when $|h_t|$ is sufficiently small. When $\tilde{\eta}_t$ rises to over $2/\lambda_1^{\mathrm{H}}(\boldsymbol{\phi}_t)$, $|h_t|$ changes from decreasing to increasing according to our approximation. But $h_t$ cannot increase indefinitely either, since $\tilde{\eta}_t$ decreases when $|h_t|$ is sufficiently large. A period finishes when $\tilde{\eta}_t < 2/\lambda_1^{\mathrm{H}}(\boldsymbol{\phi}_t)$ holds again.

In our theoretical analysis, we connect this periodic behavior with a 1-dimensional Hamiltonian system (see Appendix H.2), and show that $2h_t^2$ in each step can be approximated by its average value in the period without incurring a large error. Further calculations show that this average value is approximately $\frac{2\eta_{\mathrm{in}}}{4 + \|\nabla_\Gamma \log \lambda_1^{\mathrm{H}}(\boldsymbol{\zeta}(t\eta_{\mathrm{in}}))\|_2}$, the learning rate in the flow (3) multiplied with $\eta_{\mathrm{in}}$. We can therefore conclude that each step of $\boldsymbol{\phi}_t$ (or $\boldsymbol{\theta}_t$) tracks a time interval of $\eta_{\mathrm{in}}$ in the flow.

**Extensions.** We note that this periodic behavior is not limited to GD+WD on scale-invariant loss, since the above intuitive argument holds as long as the effective LR changes adaptively with respect to gradient change. Based on this intuition, an important notion called *Quasi-RMSprop scheduler* is proposed. For a PGD method, a learning rate scheduler is a rule for changing the effective LR in each step, and Quasi-RMSprop is a specific class of schedulers we define, including the way that the effective LR changes in GD+WD on scale-invariant loss (if viewed as PGD). Our proof is done in a unified way that works as long as the effective LR changes in each step according to a Quasi-RMSprop scheduler. As a by-product, a similar theorem can be proved for GD (without projection) on non-scale-invariant loss if the LR changes as a Quasi-RMSprop in each step. For example, we can extend our analysis to RMSprop with a scalar learning rate. See Appendix B.

## 5 Case Study: Linear Regression with Batch Normalization

In this section, we analyze the GD+WD dynamics on linear regression with Batch Normalization (BN), as a simple application of our theory. Let $\{(\boldsymbol{x}_i, y_i)\}_{i=1}^n$ be a dataset, where $\boldsymbol{x}_i \in \mathbb{R}^d$ and $y_i \in \mathbb{R}$ are inputs and regression targets. We study the over-parameterized case where $d \gg n$, and we assume that the regression targets are generated by an unknown linear model.

A classic linear model is parameterized by $(\boldsymbol{w}, b) \in \mathbb{R}^d \times \mathbb{R}$ and outputs $\boldsymbol{w}^\top \boldsymbol{x} + b$ given input $\boldsymbol{x}$, but now we add a BN to the output. More specifically, we consider a batch-normalized linear model $\Phi(\boldsymbol{x}; \boldsymbol{w}, \gamma, \beta) := \gamma \cdot \frac{\boldsymbol{w}^\top \boldsymbol{x} - \mu_1}{\sigma_1} + \beta$, where $\mu_1, \sigma_1$ are the mean and standard deviation of $\{\boldsymbol{w}^\top \boldsymbol{x}_i\}_{i=1}^n$ over the whole dataset[3], and the bias term $b$ is cancelled out due to BN. Note that $\Phi(\boldsymbol{x}; \boldsymbol{w}, \gamma, \beta)$ is still a linear function with respect to $\boldsymbol{x}$. Let $\boldsymbol{\mu}_{\mathrm{x}} \in \mathbb{R}^d$ and $\boldsymbol{\Sigma}_{\mathrm{x}} \in \mathbb{R}^{d \times d}$ be the mean and covariance of the input data $\{\boldsymbol{x}_i\}_{i=1}^n$. Then $\Phi(\boldsymbol{x}; \boldsymbol{w}, \gamma, \beta)$ can be rewritten as:

$$\Phi(\boldsymbol{x}; \boldsymbol{w}, \gamma, \beta) = \tilde{\boldsymbol{w}}^\top \boldsymbol{x} + \tilde{b}, \qquad \text{where} \quad \tilde{\boldsymbol{w}} := \gamma \boldsymbol{w} / \|\boldsymbol{w}\|_{\boldsymbol{\Sigma}_{\mathrm{x}}}, \quad \tilde{b} := \beta - \tilde{\boldsymbol{w}}^\top \boldsymbol{\mu}_{\mathrm{x}}. \tag{4}$$

No matter how $\boldsymbol{w}$ is set, the output mean and variance of $\Phi$ are always $\beta$ and $\gamma^2$. To simplify our analysis, we fix $\beta, \gamma$ to be non-trainable constants so that the mean and variance of $\Phi$'s output match with those of $\{y_i\}_{i=1}^n$, that is, we set $\beta = \mu_{\mathrm{y}}$ and $\gamma = \sigma_{\mathrm{y}}$ to be the mean and standard deviation of $y_i$ over the whole dataset. Then the training loss is $\mathcal{L}(\boldsymbol{w}) := \frac{1}{n} \sum_{i \in [n]} (\Phi(\boldsymbol{x}_i; \boldsymbol{w}, \gamma, \beta) - y_i)^2$.

**Theorem 5.1.** *In our setting of linear regression with BN, the sharpness-reduction flow $\boldsymbol{\zeta}$ defined in* (3) *converges to the solution $\boldsymbol{w}^* \in \mathbb{S}^{d-1}$ that minimizes sharpness $\lambda_1^{\mathrm{H}}(\boldsymbol{w}^*)$ on $\Gamma$, regardless of the initialization. Moreover, the coefficients $(\tilde{\boldsymbol{w}}, \tilde{b})$ associated with $\boldsymbol{w}^*$ (defined in* (4)*) are the optimal solution of the following constrained optimization problem* (M):

$$\min \quad \|\boldsymbol{w}\|_2^2 \quad s.t. \quad \boldsymbol{w}^\top \boldsymbol{x}_i + b = y_i, \quad \forall i \in [n]. \tag{M}$$

At first sight the result may appear trivial because the intent of WD is to regularize $L^2$-norm. But this is deceptive because in scale-invariant nets the regularization effect of WD is not explicit. This result also challenges conventional view of optimization. GD is usually viewed as a discretization of its continuous counterpart, gradient flow (GF), and theoretical insight for the discrete update including convergence rate and implicit bias is achieved by analyzing the continuous counterpart (See Appendix A for a list). However, GF does not have the same sharpness-reduction bias as GD. As discussed in [77], adding WD only performs a time-rescaling on the GF trajectory on scale-invariant loss, but does not change the point that GF converge to if we project the trajectory onto the unit sphere. One can easily show that GF may converge to any zero-loss solution, but no matter how small

---

[3]Note that the batch size is $n$ here as we are running full-batch GD

LR is, GD exhibits the sharpness-reduction bias towards the optimal solution of (M). To our best knowledge, this result is the first concrete example where even with arbitrarily small LR, GD can still generalize better than GF under natural settings.

## 6 Discussion

**Experimental Verification of Sharpness Reduction.** Besides Figures 1 and 2, Appendix P.1 provides additional matrix completion experiments with different data size, and Appendix P.2 provides CIFAR-10 experiments with ResNet-20. In all these experiments, we observed that GD continues to improve the test accuracy even after fitting the training set, and this phenomenon is correlated with the decreasing trend of spherical sharpness. See also Appendix P.3 for the validation for the periodic behavior we analyze in theory.

**Ablation Studies on Normalization and Weight Decay.** Our theoretical analysis crucially relies on the interplay between normalization and WD to establish the sharpness-reduction flow. We also conducted ablation studies on normalization and WD to highlight the importance of this interplay. First, if normalization is removed, the spherical sharpness becomes undefined, and we do not know if GD implicitly minimizes any sharpness measure. But even if a similar measure does exist, it cannot be strongly related to generalization, because we can verify that the test accuracy becomes very bad without normalization ($56.8\%$ on CIFAR-10, Figure 14), and continuing training after fitting the training set no longer improves test accuracy. Second, if WD is removed, the analysis in Arora et al. [7] guarantees convergence in the stable regime, and we can verify that the spherical sharpness and test accuracy stop changing when the loss is small. The final test accuracy is stuck at $66.4\%$ (Figure 15), whereas training with WD leads to $84.3\%$.

**Explaining the Progressive Sharpening and EoS Phenomena.** Cohen et al. [24] conducted extensive empirical studies on the dynamics of GD in deep learning (without weight decay), formally $\boldsymbol{w}_{t+1} \leftarrow \boldsymbol{w}_t - \hat{\eta}\nabla\tilde{\mathcal{L}}(\boldsymbol{w}_t)$. They observed the *progressive sharpening* phenomenon: $\lambda_1(\nabla^2\tilde{\mathcal{L}}(\boldsymbol{w}_t))$ tends to increase so long as it is less than $2/\hat{\eta}$. Then they observed that the training typically enters the EoS regime, which they define as a regime that (1) $\lambda_1(\nabla^2\tilde{\mathcal{L}}(\boldsymbol{w}_t))$ hovers right at, or just above $2/\hat{\eta}$; and (2) the training loss $\tilde{\mathcal{L}}(\boldsymbol{w}_t)$ goes up and down over short timescales, yet still decreases in the long-term run. A recent research trend focuses on explaining the progressive sharpening and EoS phenomena [1, 91, 8, 18]. Our work corresponds to an important special case where $\tilde{\mathcal{L}}(\boldsymbol{w})$ is a scale-invariant loss with $L^2$-regularization, namely $\mathcal{L}(\boldsymbol{w}) + \frac{\hat{\lambda}}{2}\|\boldsymbol{w}\|_2^2$. By analyzing the interplay between normalization and WD, the first part of our results (Section 4.1) attributes progressive sharpening to norm change, and the second part (Section 4.2) justifies in theory that the training can make progress in the EoS regime. See Appendix G.3 for more discussion.

## 7 Conclusions and Future Work

We exhibited settings where gradient descent has an implicit bias to reduce spherical sharpness in training neural nets with normalization layers and weight decay, and we verified experimentally this sharpness-reduction bias predicted by our theorem as well as its generalization benefit on CIFAR-10.

Our theoretical analysis applies to dynamics around a minimizer manifold and requires a small (but finite) learning rate so that we can show that the parameter oscillates locally and approximately tracks a sharpness-reduction flow. We note that in practice a decrease in spherical sharpness is observed even with moderate LR and even before getting close to a minimizer manifold. Explaining these phenomena is left for future work. Now we list some other future directions. The first is to generalize our results to SGD, where the sharpness measure may not be the spherical sharpness and could depend on the structure of gradient noise. Second, to understand the benefit of reducing spherical sharpness on specific tasks, e.g., why does reducing spherical sharpness encourage low-rank on matrix completion with BN (Figure 1)? Third, to study sharpness-reduction bias for neural net architectures that are not scale-invariant on all parameters (e.g., with certain unnormalized layers).

## Acknowledgements

This work is funded by NSF, ONR, Simons Foundation, DARPA and SRC. ZL is also supported by Microsoft Research PhD Fellowship.

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
