# OpenReview forum: "Understanding the Generalization Benefit of Normalization Layers: Sharpness Reduction"
_NeurIPS.cc/2022/Conference — NeurIPS 2022 Accept_

### Official Review · Reviewer_pjae · 2022-07-11

**Rating:** 8
**Confidence:** 3
**Soundness:** 4 excellent
**Presentation:** 2 fair
**Contribution:** 4 excellent

**Summary:**

This paper studies the dynamics of sharpness for networks with normalization layers (invariant losses) that are trained using gradient descent with weight decay. The authors define spherical sharpness to account for the scale-invariance of the loss function. They first show theoretically that when neural network parameters get close to a minimizer, after some time, they would enter the "edge of stability" regime where both the learning rate and the network parameters would oscillate. Next, it is theoretically shown that at this edge of stability, the network parameters converge to a sharpness-reduction flow with a period-2 oscillation.  Finally, the theoretical results are showcased for a few experimental settings.

**Questions:**

1- It is not clear why the combination of BN and WD is interesting to study? It seems a bit random.

2- Why are so many jumps in the train/test accuracy curves of Figure 2? What explains the drops in the training accuracy to below 100%? In typical experimental settings (training a VGG on CIFAR-10) such oscilations don't occur.

3- How are Figures 1 and 2 showing that "settings where normalization provably induces an implicit bias to persistently reduce sharpness in training, which we call the sharpness-reduction bias". It is not clear why the reduction in sharpness is linked to BN?




**Limitations:**

The limitations of the work have been mentioned in the conclusion section.

**Strengths And Weaknesses:**

Strengths:

The problem (why normalization helps generalization) is of interest to the community. The theoretical contributions are rather complete and indeed provide a reasonable answer to the problem. The technical contribution with regards to showing the sharpness reduction property for gradient descent (without any stochastic noise) is solid. The practicality of the assumptions has also been considered (lines 209-214). There are even many results that are mentioned mainly in the appendix.

Weaknesses:

The main point missing from the paper is the motivation behind the particular choice of the setting (BN+WD). Perhaps an empirical example would be nice to motivate this; first showing without BN there is no decrease in sharpness, but then after adding normalization layers there is.

I would suggest a proofread for the paper to fix typos and rewrite some sentences to make them more clear.

---

> ### Author Response · Authors · 2022-08-02
> **Response**
>
> We thank the reviewer for the appreciation of our theoretical contribution. We have fixed several typos and grammar issues in our new revision.
>
> **Q1: (Major Weakness) Why is the combination of BN and WD interesting to study?**
>
> 1. Normalization + WD is one of the most common and successful regimes in practice. Nearly all SOTA neural nets (including ResNets, DenseNets and Transformers) have normalization layers and are trained with WD.
> 2. This training regime is of particular interest because GD can exhibit unusual optimization behaviors that lead to implicit regularization. Traditional optimization theory asserts that a neural net should stop improving test accuracy after convergence, but in our intriguing experiments on CIFAR-10 with normalization + WD, GD continues to improve the test accuracy even after reaching 100% training accuracy (see Figure 2 and Appendix P.2 for more).
> 3. The role of normalization and WD is crucial because when either of them is removed, the above phenomenon does not occur. See our general comments on ablations studies.
>
> **Q2: Why are so many jumps in the train/test accuracy curves of Figure 2? In typical experimental settings such oscillations don't occur.**
>
> 1. The training dynamics may exhibit instability in presence of normalization and WD, which is a known phenomenon that is studied empirically by [Lobacheva et al., (2021)](https://openreview.net/forum?id=B6uDDaDoW4a).
> 2. The training accuracy should always be 100% in the regime we analyze theoretically. However, our theory requires the LR (or the product of LR and WD) to be sufficiently small, and the time length that the dynamics can be tracked with the sharpness-reduction flow depends on LR. In our experiments with LR = 0.1, the sharpness-reduction flow may lose track of the GD+WD dynamics when the training time is long enough.
> 3. Although the flow may lose track of the discrete dynamics and the training can exhibit instability, our theory successfully predicts that the spherical sharpness has an overall tendency to decrease.
>
> **Q3: “[we] exhibit settings where normalization provably induces an implicit bias to persistently reduce sharpness in training … See Figures 1 and 2 …” How are Figures 1 and 2 showing that?**
>
> Sorry for the confusion. That sentence does not precisely describe what Figures 1 and 2 can show. We have rewritten that sentence as the following:
>
> “[we] exhibit settings where gradient descent persistently reduces sharpness in training normalized nets with WD … See Figures 1 and 2 …”
>
> As mentioned in our general comments, we have also added ablation studies of normalization in Appendix P.4.

---

### Official Review · Reviewer_hMQT · 2022-07-11

**Rating:** 6
**Confidence:** 1
**Soundness:** 3 good
**Presentation:** 3 good
**Contribution:** 2 fair

**Summary:**

The authors presented analysis that Gradient Descent (GD) reduce spherical sharpness in training neural networks with normalization layers.
Their analysis is done by going through the trajectory of GD, from the stable regime into the Edge of Stability (EoS) regime.
In the EoS regime, the trajectory oscillates around the local minimizer manifold, and moves toward the a direction that reduces spherical sharpness.
In that regime the author shows, through inductive proof on two EoS condition, that a periodic behavior moves the gradient through until it minimizes the error.
The authors experimentally verified their findings on CIFAR-10 with batch-normalized, scale-invariant VGG-11 and ReNet-20, using Swish activation function.


**Questions:**

Typo, Grammar, clarification etc.:
Pg.3, ln. 91: generalizaiton —> generalization
Pg.3, ln. 95: with different with different —> with different
Pg.3, ln. 201: moves towards —> moves toward


**Limitations:**

The paper is theoretical in nature. Any negative societal impact would have to come from, at the current stage, a hard-to-foresee application of the research.

**Strengths And Weaknesses:**

As more of a user of deep neural networks (both Computer Vision and NLU), I feel that the paper can be inaccessible to the general AI community. For one quite a large part of the proof is in the appendix, most reader will just take it to heart that things will work out. However, through that process, reader may lose grounded-ness of what should be kept tracked.

I would say, making better use of the neural network results of the optimization at certain stages may help. That is to show what more is needed to be learned (which images went from misclassified to classified) and how batch normalization allows for that to happen.

---

> ### Author Response · Authors · 2022-08-02
> **Response**
>
> We thank the reviewer for appreciating our work and pointing out the typos. We have fixed several typos and grammar issues in our new revision.
>
> **Q1: (paraphrase) Readers may not be able to know what needs to be learned from the analysis, because a large part of the proof is in the appendix.**
>
> A proof sketch is given in Section 4.2.3, which precisely describes the main insights in our analysis. Figure 3 gives a plot that shows the dynamics intuitively in a simple case (see also the text at the beginning of Section 4.2). We will continue to polish the paper and will provide a better overview in the next revision (the final version will allow an extra page).
>
> **Q2: (paraphrase) Include examples of which images go from misclassified to classified, and how improved generalization due to BN allows for that to happen.**
>
> Thanks for the interesting question. Today’s generalization theory is not strong enough to give such a fine-grained explanation, but it is definitely an interesting future direction.

---

> > ### Comment · Reviewer_hMQT · 2022-08-07
> > **Thank You For All Your Responses**
> >
> > No further questions from me. I will keep my score as is.

---

### Official Review · Reviewer_1MUm · 2022-07-12

**Rating:** 6
**Confidence:** 3
**Soundness:** 3 good
**Presentation:** 3 good
**Contribution:** 3 good

**Summary:**

This paper concerns on the optimization aspect when training deep neural networks (DNNs) with normalization techniques. Normalized DNNs are prevalent and often have high performance in practice. However, little is theoretically known. This paper analyzes the role of using normalization and weight decay when training by gradient descent (GD). This paper shows that normalization + weight decay can help GD move to flat region, can impose an implicit bias to reduce sharpness. The authors find that when GD reach the manifold of minimizers, it can move to the Edge of Stability regime. Once in this regime, we can move to a less sharp region on the manifold when using more GD steps. Some experiments on VGG-11, ResNet-20, matrix completion, and linear regression using Batch normalization further support their theory.

**Questions:**

- Normalization methods often have their own trainable parameters. Those trainable params are important to DNNs' performance in practice. Why this paper does not take those params into consideration? Ignoring those params may not provide correct understanding of normalized DNNs.
- There is a high fluctuation after 47k training steps in Figure 2. It suggests that when moving out of minimizer manifold, the training loss will be extremely high. Does it indicate that the manifold of minimizers disconnect with the outside or there are some "hole" around that manifold?


**Limitations:**

The authors discuss some limitations of this work. I encourage the authors to state clearly the scope of this paper and redesign their experiments.

**Strengths And Weaknesses:**

Strengths:
- This paper provides an interesting view about training normalized DNNs by GD, adding to our knowledge about the behavior of practical DNNs. After reaching the manifold of minimizers, it can move to the Edge of Stability regime, and continue moving to a less sharp region. There is close connection between flat minima and generalization.
- Some experiments on some DNNs, matrix completion, and linear regression are done to further support their theory.

Weaknesses:
- Despite of the main focus on normalization, this paper uses only "scale-invariant" property to derive their results. This property appears in some methods such as layer norm, instance norm, group norm. However, some other normalization methods (e.g., batch norm - BN) do not ensure scale-invariant loss. The reason is that BN computes the variance of an input signal from the whole training data, and therefore an BN operation is data-dependent and will be dynamic over the training course. This fact violates the assumption about scale-invariant loss in this paper.
- Due to the above reason, the scope of this paper is unclear and the theory may mismatch their experiments. BN seems not to follow their theory, but the experiments use only BN. The authors should redesign their experiments and narrow down the scope.

Minor: "reduce the sharpness of the loss" may not reflect the true nature. A loss function is often fixed before training. Therefore, one may not reduce its sharpness when minimizing the loss. Instead, GD may move from a sharp point to another less sharp point.

======
After reading the authors' response, I now see no more mismatch. I hence slightly increase my score.

---

> ### Author Response · Authors · 2022-08-02
> **Response**
>
> We thank the reviewer for appreciating our theoretical analysis and empirical validation.
>
> **Major point: Experiments are about BN but it does not ensure scale-invariance.** The reviewer wrote that “BN computes the variance of an input signal from the whole training data” and “This fact violates the assumption about scale-invariant loss in this paper.”
>
> **Our response:** BN does ensure scale-invariance. As noted in the original BN paper [(Ioffe & Szegedy, 2015)](https://arxiv.org/abs/1502.03167), BN indeed makes the loss function invariant to rescaling of weights in linear layers, and thus the loss is scale-invariant with respect to these weights (see Section 3.3 herein). The reason is that the variance is computed on the current batch. If the weight $w$ is changed to $cw$ ($c > 0$ is a factor), then the input signal to BN will be multiplied by $c$ and the variance will be multiplied by $c^2$. According to the formula of BN, the output of BN (roughly input / sqrt(variance)) should be unaffected.
>
> **Q1: Why does this paper not take the non-scale-invariant params into consideration? E.g., the trainable params in normalization layers.**
> 1. To highlight the effect of scale-invariance led by normalization, we made minor changes to VGG and ResNet so that they are scale-invariant wrt all parameters. See Appendix Q.
> 2. With ReLU activation, the loss function can also be scale-invariant wrt the trainable parameters in normalization layers, if this normalization is not the last one. This is because ReLU is 1-homogeneous, and the output of ReLU is subsequently processed by a linear layer and a normalization layer. See also Appendix E of [(Li & Arora, 2020)](https://openreview.net/pdf?id=rJg8TeSFDH) for detailed discussion.
>
> **Q2: There is a high fluctuation in Figure 2. Does it indicate that the manifold of minimizers disconnects with the outside or there are some "holes" around that manifold?**
> 1. The structure of the manifold can depend on neural net architecture and training data. We are unable to analyze its structure in detail due to the complexity in math.
> 2. The high fluctuation suggests that the sharpness-reduction flow sometimes loses track of GD during training if the LR is not small enough. See also our response to Q2 of Reviewer pjae.

---

> > ### Comment · Reviewer_1MUm · 2022-08-06
> > **The mismatch and fluctuation**
> >
> > I thank the authors for pointing out my misunderstanding about BN. I now see no more mismatch, and adjust my score.
> > One thing still in my mind is the high fluctuation of the training loss in Figure 2. This behavior should need more investigation.

---

### Official Review · Reviewer_CDHW · 2022-07-13

**Rating:** 6
**Confidence:** 3
**Soundness:** 4 excellent
**Presentation:** 3 good
**Contribution:** 3 good

**Summary:**

This paper shows that normalization along with weight decay biases full-batch gradient descent with finite step sizes towards flat solutions.

**Questions:**

N/A

**Limitations:**

The authors do discuss their assumptions in detail, and they do mention the limitations of their work at understanding broader deep learning phenomena.

**Strengths And Weaknesses:**

Strengths and Contributions:

-The paper’s organization is great, and it’s easy to follow.

-This paper explains the EoS phenomenon theoretically, which to my knowledge was previously a puzzling unexplained observation.

-The paper does a reasonable job of providing intuition behind the theory.


Weaknesses

-Throughout the paper, starting in the first sentence of the intro, grammatical errors make the paper a bit hard to read, but I was still able to understand everything after re-reading some sentences a couple times.  Not a huge deal but definitely worth fixing before publication.

-Empirically, the benefits of weight decay are unclear.

-Other works have found that full-batch GD finds sharper mins than SGD in the absence of, for example, gradient clipping.  The difference between sharp/flat properties of full-batch and stochastic training calls into question exactly what this work can tell us about the dynamics of training routines people actually use, which are nearly always stochastic.  It would be interesting to connect your findings to the effects of stochasticity on training.

-The empirical validation is very sparse.  I realize that there has already been a lot of work on the sharp/flat hypothesis and normalization layers from an empirical perspective, but ablations on the assumptions of the theory (i.e. experiments to see if the predictions of the theory still hold empirically under weaker assumptions) would be interesting.

-Minor point: the scale invariance problem for flatness measurements like the spectral norm of the Hessian is a very special (and particularly coarse) case of a broader space of reparameterization problems by which different parameter settings may yield the same function but not the same flatness.  A number of stronger invariant metrics have been proposed, for example invariant to filter-wise rescaling, but it would also be interesting to show function-space properties (e.g. classifier margins, etc) rather than parameter-space ones.

---

> ### Author Response · Authors · 2022-08-02
> **Response**
>
> We thank the reviewer for appreciating our theoretical explanation for EoS in normalized nets. We have fixed several typos and grammar issues in our new revision.
>
> **Q1: “Empirically, the benefits of weight decay are unclear.”**
>
> Empirically, adding WD to normalized nets is beneficial to generalization. [Zhang et al., (2018)](https://arxiv.org/abs/1810.12281) demonstrated significant performance gaps in various settings (see, e.g., Figure 1). [Liu et al., (2019)](https://arxiv.org/pdf/1906.02613.pdf) did an ablation study showing that adding WD helps vanilla SGD escape from global minima that generalize worse (see Table 2).
>
> **Q2: (paraphrase) “The result is about GD; gives no insight into practice, where everybody uses SGD. “**
>
> Analyzing SGD is left for future work. But an analysis for GD (which turns out to be quite complicated!) is a necessary first step toward understanding SGD, since any theoretical analysis of SGD should automatically imply one for full-batch GD when the batch size is large.
>
> **Q3: “Ablations on the assumptions of the theory would be interesting.”**
>
> Thanks for the suggestion. We have included more experiments in Appendix P (Change #1).
>
> **Minor Point: (paraphrase) “The spherical sharpness only captures the scale-invariance, but real-world neural nets can have many other symmetries.”**
>
> Our primary goal is not to identify the correct sharpness metric under various symmetries, but to understand how GD can find flat solutions for normalized nets. Although there are many other sharpness metrics that take care of more symmetries, it is unclear whether GD/SGD is implicitly reducing them.

---

> > ### Comment · Reviewer_CDHW · 2022-08-06
> > **Thanks for your response**
> >
> > I am somewhat satisfied by your response, but I do want to touch on a couple points:
> >
> > Given that neural networks, for example in computer vision, typically outperform their competitors by a very large margin with or without weight decay and normalizers, and in fact some SOTA systems do not have the scale invariance property, it still remains unclear just how much behavior of neural networks the theory in this paper can explain.  That said, my original paper score reflected that I think this paper covers an interesting direction that some members of the NeurIPS audience would be interested in, so I vote for acceptance.
> >
> > **"Although there are many other sharpness metrics that take care of more symmetries, it is unclear whether GD/SGD is implicitly reducing them"** - This is true, but it is in fact a problem that gets in the way of drawing connections between your work and any conclusions about why neural networks generalize well because the flatness metric you adopt may not in fact be the one relevant for generalization, and one obvious reason why this may be this case is that your metric does not take into account a variety of symmetries in neural networks.

---

### Official Review · Reviewer_qMdw · 2022-07-16

**Rating:** 6
**Confidence:** 5
**Soundness:** 3 good
**Presentation:** 3 good
**Contribution:** 3 good

**Summary:**

The current paper argues that benefits of BatchNorm in regards to generalization have been under-discussed and tries to develop a novel analysis for the same. To this end, the authors prove that under weight decay and a scale-invariant loss, as induced if there is a BatchNorm corresponding to all parameters, the model's training functions in two regimes: stable and edge-of-stable (EoS), which arises because the use of weight decay forces parameter norm to increase. In the EoS regime, under assumptions such as existence of a manifold of minimizers, it is shown that scale invariance will force the model to follow a "sharpness-reduction" flow, i.e., a gradient flow under which the sought solution is trying to minimize sharpness. Sharpness is defined by taking scale-invariance into account.

**Questions:**

See weaknesses.

**Limitations:**

Primarily, there is scope for improvements in experiments and discussion on relevant related works.

**Strengths And Weaknesses:**

### Strengths
I think the analysis in this paper is strong and the proposed tools, though extensions of prior works to some extent, can be of interest beyond the paper's goal. The overall story is very well laid and I really enjoyed reading the paper.


### Weaknesses

I think the paper has few primary weaknesses in its current form and they should be easy to address. I strongly encourage the authors to do the same and am happy to raise my score accordingly.

* Related work: While the related work is decently broad, I think a few highly relevant papers have been missed. Specifically, Luo et al. (https://arxiv.org/abs/1809.00846) developed an objective that approximates BatchNorm's regularization effects. It would be important to cite the paper and compare if its results relate with the proposed sharpness reduction flow (could they be similar in some settings?). Second, Tanaka and Kunin (https://arxiv.org/abs/2105.02716) developed a detailed relationship between models trained using SGD under normalization layers and adaptive gradient methods such as RMSprop. The authors claim their proposed link is the first, but this paper has a similar relationship derived via analyzing symmetry breaking of scale invariant losses under fiinite learning rate. It would be important to highlight how the authors' results differ.

* Experiments: While I enjoyed the analysis, I think the experiments lack some ablative baselines demonstrating how lack of BatchNorm would have not yielded similar generalization or similar sharpness reduction. For example, in Figure 2 with CIFAR-10 results, one can train the model without BatchNorm and achieve similar or better test accuracy using the initialization strategy proposed by Daneshmand et al. (see section 4 of https://arxiv.org/abs/2003.01652). Would one see a similar reduction in sharpness? In fact, Daneshmand et al. argue in their work that BatchNorm's effectiveness cannot be explained via scale invariance and as long as one gets a rank-maximizing signal flow at initialization, good generalization is achieved. Then, the experiment suggested above will help understand whether the current paper's position is correct.

* Extension to other layers: I think it is important to include discussion on why similar results on sharpness reduction will not be seen with other normalization layers. Technically, most normalization layers have scale invariance at some granularity (e.g., channel-level in InstanceNorm or layer-level in LayerNorm). Several of those layers are unable to match BatchNorm in its effectiveness, but some of them do when they have similar signal propagation characteristics, as shown by Lubana et al. (https://arxiv.org/abs/2106.05956).

---

> ### Author Response · Authors · 2022-08-02
> **Response**
>
> We thank the reviewer for the appreciation of our theoretical analysis and for pointing out the relevant papers. We will include a more complete discussion on how people understand normalization in the next revision.
>
> **A general comment.** Our paper shows that *all* forms of normalization layers (which lead to scale-invariance) have a tendency to reduce sharpness at the late phase of training when loss becomes small. We do not claim that this is the *only* explanation for why normalization helps with good generalization, and it is possible that some forms of normalization are better than others due to other reasons.
>
> **Q1: (paraphrase) If the sharpness-reduction bias can also be seen with other forms of normalization, then this paper gives no insight into why several of them are unable to match BN in its effectiveness.**
>
> We do not claim that the sharpness reduction is the *only* mechanism that can lead to better generalization. It is generally believed that sharpness is only one of many factors that promote generalization (see e.g., experiments in [(Jiang et al. 2019)](https://arxiv.org/abs/1912.02178)). The mechanisms identified in other papers (e.g., the papers that the reviewer mentions: Daneshmand et al., Lubana et al., Luo et al.) could also be contributing to good generalization.
>
> **Q2: (paraphrase) Past work (e.g., Daneshmand et al.) has shown that BN is not needed and can be replaced by changing initialization. Will one see a similar reduction in sharpness?**
>
> 1. Normalization may sometimes not be needed, but in many settings (including transformers) we don’t know how to remove it without greatly hurting performance.
> 2. It is not easy to see what would happen in Daneshmand et al.’s setting. As mentioned in our general comments, spherical sharpness (Def. 1.1) is meaningful only for normalized nets. For unnormalized nets, we do not know any sharpness measure that behaves like in our paper (monotone decreasing during the last phase of training and related to generalization in unnormalized nets).
>
> **Q3: Relationship to Luo et al. (2019).**
>
> Luo et al. (2019) studied the regularization effects of the last-layer BN due to sampling noise, while in our setting we consider full-batch GD. We do not see the connection between their regularizer and our sharpness reduction flow.
>
> **Q4: Tanaka & Kunin (2021) developed a detailed relationship between SGD + normalization and adaptive gradient methods, but the authors claim that their proposed link is the first.**
>
> 1. The main theme of our paper is to study the generalization benefits of normalization layers via analyzing the implicit bias of GD. We do not claim that we are the first to study the relationship between GD + normalization and adaptive gradient methods.
> 2. Tanaka & Kunin’s derivation is for the continuous-time approximation of the momentum method (a second-order ODE), not for SGD.
> 3. We show a similar connection between GD on scale-invariant loss and RMSprop (Theorem B.7). The differences are: (1) ours is for vanilla GD while theirs is for momentum; (2) ours is for discrete dynamics with finite LR while theirs is for continuous ones.

---

> > ### Comment · Reviewer_qMdw · 2022-08-04
> > **Response to rebuttals**
> >
> > Thank you to the authors for their response. The questions I posed have been addressed and I see the paper has some updated discussion around closely relevant works now (though I would recommend discussing Tanaka and Kunin (2021) in the main paper because of its relevance to RMSprop analysis).
> >
> > I will keep my score the same because the score description best describes my assessment of the paper: "Technically solid, moderate-to-high impact paper, with no major concerns".

---

> > > ### Author Response · Authors · 2022-08-07
> > > **About the related works**
> > >
> > > We thank the reviewer for the response. Below we would like to explain more about our efforts in response to the reviewer's request to add more discussion on the related works.
> > >
> > > 1. We have cited all the papers the reviewer mentioned, and we promise to include a complete discussion on how people understand normalization (including but not restricting to the analyses based on scale-invariance) in the next revision.
> > > 2. We have added a discussion on Tanaka & Kunin's work since it is related to our connection between scale-invariance and RMSprop. In the current version, it only appears in the appendix. We didn't discuss Tanaka & Kunin's work in the main paper because we didn't explicitly write out this connection to RMSprop in the main paper, either.
> > > 3. In the final version of our paper, an extra page will be allowed, and we will consider explaining more about this connection in the main paper. If so, we will move the discussion about Tanaka & Kunin's work from the appendix to the main paper as well.
> > > 4. We would like to clarify again that the focus of our paper is not to establish the connection between scale-invariance and RMSprop, but to use this connection (as one of the first steps in our proof) to analyze the generalization benefits of normalization. This requires many more proof insights that do not appear in previous works (Section 4.2.3 and Appendix H).

---

### Author Response · Authors · 2022-08-02
**Authors’ General Comments**

We thank all the reviewers for their appreciation of our result, i.e., a theoretical analysis showing that GD implicitly minimizes spherical sharpness on normalized nets.

### New Revision Uploaded

**Change #1.** More experiments are added to Appendix P, including ablation studies on normalization and WD.

**Change #2.** We have fixed several typos and grammar issues.

### Ablation Studies
As more than one reviewer asked to see ablation studies of normalization and WD to better understand the effect highlighted in our theory, we have added such experiments to Appendix P.

**Training with normalization and WD.** In our intriguing experiments on CIFAR-10 with normalization + WD, GD continues to improve the test accuracy even after reaching 100% training accuracy (see Figure 2 and Appendix P.2 for more), and the overall tendency of spherical sharpness is to decrease.

**Ablation study of normalization.**
1. In a CIFAR-10 + VGG-11 experiment with the same setting as Figure 2 but without normalization, the test accuracy is ~56% and we do not see the phenomenon that test accuracy keeps increasing after the training accuracy reaches 100%. See Figure 13 in our new revision.
2. Spherical sharpness (Def. 1.1) is defined for normalized nets. For unnormalized nets, we do not know any sharpness measure that behaves like spherical sharpness (monotone decreasing during the last phase of training and related to generalization in unnormalized nets). Even if GD does implicitly reduce a similar sharpness measure, such a sharpness measure cannot be strongly related to generalization because the test accuracy does not increase accordingly.

**Ablation study of WD.** In a CIFAR-10 + VGG-11 experiment with the same setting as Figure 2 but without WD, the spherical sharpness does not change much after a few initial steps, and the test accuracy is stuck at 66.4%, whereas training with WD leads to 84.3%. See Figure 14 in our new revision.

---

### Meta-Review · Area_Chair_RcMX · 2022-08-21

**Recommendation:** Accept
**Confidence:** Certain

**Metareview:**


This papers demonstrates how normalization with weight decay forces GD to converge to flater solutions.
Overall, the reviewers find that the paper does bring some new insights, perhaps with a limited impact but still valuable to the community. There were also a few concerns left after the discussion period, e.g.
- Dicussion of prior work is rather superficial. Some key works are missing or not discussed in appropriate depth
- Limited empirical evaluation

Given the discussion with the reviewers, the first concern seems easilly fixable in the camera-ready version. Regarding the second concern, the paper does however compensate with a solid technical contribution. I therefore recommend acceptance but I strongly encourage the authors to update their manuscript to address the above concerns. The writing quality is sometimes poor and should be improved in the final version.

**Award:**

No

---

### Decision · Program_Chairs · 2022-09-14

Accept